# Top-down inference in an early visual cortex inspired hierarchical Variational Autoencoder

**Ferenc Csikor**[*]
Department of Computational Sciences
Wigner Research Centre for Physics
Budapest, Hungary
`ferenc.csikor@gmail.com`

**Balázs Meszéna**[*]
Department of Computational Sciences
Wigner Research Centre for Physics
Budapest, Hungary
`meszenab@gmail.com`

**Bence Szabó**
Department of Computational Sciences
Wigner Research Centre for Physics
Budapest, Hungary
`bszabo96@gmail.com`

**Gergő Orbán**
Department of Computational Sciences
Wigner Research Centre for Physics
Budapest, Hungary
`orgergo@gmail.com`

## Abstract

Interpreting computations in the visual cortex as learning and inference in a generative model of the environment has received wide support both in neuroscience and cognitive science. However, hierarchical computations, a hallmark of visual cortical processing, have remained impervious for generative models because of the lack of adequate tools to address it. Here, we capitalize on advances in Variational Autoencoders (VAEs) to investigate the early visual cortex with sparse-coding two-layer hierarchical VAEs trained on natural images. We show that representations similar to those found in the primary and secondary visual cortices naturally emerge under mild inductive biases. That is, the high-level latent space represents texture-like patterns reminiscent of the secondary visual cortex. We show that a neuroscience-inspired choice of the recognition model is important for learning noise correlations, performing image inpainting, and detecting illusory edges. We argue that top-down interactions, a key feature of biological vision, born out naturally from hierarchical inference. We also demonstrate that model predictions are in line with existing V1 measurements in macaques with regard to noise correlations and illusory contour stimuli.

## 1 Introduction

Interpreting visual perception as inference in an unsupervised generative model is a key concept in neuroscience [45, 14] and is currently an active research area [11]. Specifically, using *generative* models of natural images to understand the response statistics of neurons in the visual cortex of mammals has proven to be a lucrative approach. Linear (or close to linear) models have been instructive in accounting for a wide spectrum of properties in the response characteristics of neurons found in the primary visual cortex (V1). This includes the structure of receptive fields [32, 4, 24, 23], the properties of extra-classical receptive fields [38], or response variability [22, 33, 12, 13]. However, going beyond linear models (and V1) has been largely hampered by limits on the capabilities of machine learning tools to perform learning and inference in nonlinear generative models. Unlike

---

[*]equal contributions

4th Workshop on Shared Visual Representations in Human and Machine Visual Intelligence (SVRHM) at the Neural Information Processing Systems (NeurIPS) conference 2022. New Orleans.

discriminative models, where detailed mapping exists between multiple cortical areas and latent layers [44], generative models rarely describe different hierarchical regions [21].

Recently, nonlinear generative models have attracted considerable attention [26, 37, 17, 27]. Variational Autoencoders (VAEs) are a flexible class of deep generative models that use deep neural networks to parameterize highly nonlinear computations necessary to discover nonlinear and potentially disentangled latent features, as well as delicate invariances characteristic of higher-level visual areas [10, 48, 7, 20, 47]. Identifying higher-level cortical representations with VAE representations in higher area is especially appealing, since inference in ($\beta$-)VAEs is known to correspond to compression [1, 20]. Surprisingly, the application of VAEs to neural computations has been limited to a single layer [20, 3].

The visual cortex of primates and other mammals is characterized by a hierarchy of processing stages. Consequently, establishing links between deep generative models and visual cortical computations requires hierarchical versions of VAEs (hVAEs). Recently, considerable advances have been made in the construction of effective hVAE architectures [30, 41, 46, 9, 39, 19]. Despite recent successes in generating high-quality images with hVAEs, their learned representations are less studied [8].

We investigate the properties of representations and inference in an hVAEs. We are especially interested in these probabilistic generative models because neuronal correlations and top-down phenomena (such as illusory contour detection) can be addressed with them. We use a VAE with two latent layers that can be related to the primary and secondary visual cortices (V1 and V2). In an hVAE, architectural choices concern both the generative and the recognition components. One of such choices is relying on skip connections in the generative model, which link higher layers of latent variables to observed variables to stabilize learning [9]. However, it is less well motivated from an interpretable representation learning and neuroscience perspective than a Markovian generative model where there is a clear hierarchy between latent layers. In contrast, a recognition model that features skip connection also contains a top-down component, which is well motivated by neuroscience.

In this paper, we first develop a biologically inspired form of hVAE, which we call TDVAE (Top-down VAE) together with some alternative architectures (SkipVAE, ChainVAE and a single layer LinearVAE). We find that hVAE architectures are capable of reproducing some key properties of representations emerging in V1 and V2 of the visual cortex. Further, we find that architectures that feature top-down influences in their recognition model give rise to a richer representation, such that specific information that is not present in mean activations becomes linearly decodable from posterior correlations. In the literature, posterior correlations have been shown to be related to neuronal noise correlation (NC) [22, 33, 12]. We link this result with earlier findings in monkey electrophysiology, where the stimulus specificity of NC was shown to change when stimulus statistics were manipulated [2]. Finally, we investigate top-down contributions to contextual effects: inpainting of image pixels and illusory percepts in the latent space. Similarly to activity in V1 neurons [28], we find that illusory edges can appear in latent responses, and this is enhanced by top-down connection. A similar contribution is identified in image inpainting where masked images are 'corrected', relying on information delivered by top-down connections.

## 2 Methods

We study two-layer hierarchical latent variable generative models that learn the joint distribution of observations and latent variables in the form $p_\theta(\mathbf{x}, \mathbf{z}_1, \mathbf{z}_2) = p_\theta(\mathbf{x} \mid \mathbf{z}_1, \mathbf{z}_2) \cdot p_\theta(\mathbf{z}_1 \mid \mathbf{z}_2) \cdot p_\theta(\mathbf{z}_2)$. To make inference tractable, we train VAEs [26, 27], where the generative model is supplemented by a recognition model. This establishes a variational approximation ($q_\Phi(\mathbf{z_1}, \mathbf{z_2} \mid \mathbf{x})$) of the true posterior distribution ($p_\theta(\mathbf{z_1}, \mathbf{z_2} \mid \mathbf{x})$). In general, the ELBO for a two-layer VAE can be written as

$$\text{ELBO}(\mathbf{x}, \theta, \Phi) = \text{E}_{q_\Phi(\mathbf{z_1}, \mathbf{z_2} \mid \mathbf{x})}[\log p_\theta(\mathbf{x} \mid \mathbf{z_1}, \mathbf{z_2})] - \text{KL}[q_\Phi(\mathbf{z_1}, \mathbf{z_2} \mid \mathbf{x}) \,\|\, p_\theta(\mathbf{z_1}, \mathbf{z_2})]. \quad (1)$$

In our main model (TDVAE) we use a Markovian structure where $p_\theta(\mathbf{x} \mid \mathbf{z_1}, \mathbf{z_2}) = p_\theta(\mathbf{x} \mid \mathbf{z_1}) \cdot p_\theta(\mathbf{z_1} \mid \mathbf{z_2})$. We will see that the representation learned by such a Markovian generative model is more interpretable as the low-level and higher-level (more semantic) latent variables decouple. When dealing with a variational approximation of a hierarchical generative model, there are also multiple choices to consider on how to factorize the recognition model in terms of $\mathbf{z_1}$ and $\mathbf{z_2}$. In the top-down case, we have $q_\Phi(\mathbf{z_1}, \mathbf{z_2} \mid \mathbf{x})^{TD} = q_\Phi(\mathbf{z_2} \mid \mathbf{x}, \mathbf{z_1}) \cdot q_\Phi(\mathbf{z_1} \mid \mathbf{x})$ (as in TDVAE). In the bottom-up version, we have $q_\Phi(\mathbf{z_1}, \mathbf{z_2} \mid \mathbf{x})^{BU} = q_\Phi(\mathbf{z_2} \mid \mathbf{z_1}) \cdot q_\Phi(\mathbf{z_1} \mid \mathbf{x})$ (assuming a Markovian

generative model). The mathematical motivation for these form of factorizations is that they mimic the statistical properties of the true posterior (see A.1.1).

The components of the generative and recognition models are parameterised through neural networks with softplus activation function. We use fully connected networks instead of CNNs to avoid the indirect effects of CNNs on emerging representations. We implemented standard neuroscience-inspired biases at the level of $\mathbf{z}_1$ inspired by models of V1 activity[32]. Consequently, we choose to implement the lower layer as an overcomplete ($dim(\mathbf{z}_1) > dim(x)$) layer with a sparse (Laplace) prior, having a linear generative relationship to observations $p_\theta(\mathbf{x}\,|\,\mathbf{z}_1) = \mathcal{N}(\mathbf{x}; \mathbf{A}\mathbf{z}_1, \mathbf{I})$.

In Fig. 1(a)-(d) we summarize the high-level architectures that appear in this paper. For easier reference, we use the following terminology throughout the paper: a) LinearVAE is a single-layer VAE with linear generative (and nonlinear recognition) model (this setup is identical to the one in [3]); b) TDVAE is a two-layer VAE with Markovian generative model and top-down factorization of the variational posterior; c) ChainVAE is a two-layer VAE with Markovian generative model and bottom-up factorization of the posterior; and d) SkipVAE is a two-layer VAE with non-Markovian generative model and top-down factorization of the posterior. For more details on the derivation of the explicit objective functions and model architectures, see A.1.

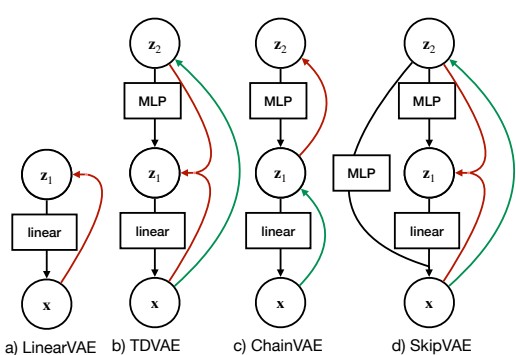

Figure 1: (a)-(d) The generative models used in the paper and schematic representations of the recognition models. LinearVAE is a single-layer model, while the rest have two layers.

Importantly, to match the conditions to which a biological system is adapted, we use natural images for training. Inspired by the sensitivities of neurons in V2 [16, 47, 15] (a cortical area beyond V1 in the processing hierarchy) to explore emerging representations, we also use synthetic texture images for testing. Training was performed on $40 \times 40$ pixel whitened natural image patches. For testing purposes we also used synthetic texture images which can be categorized into five distinct families. Textures have a particular appeal, since they are characterized by nonlinear sufficient statistics [35, 40] for which biological systems show high sensitivity to. More details about the preprocessing of natural images and texture generation can be found in A.2.

## 3   Experiments

**Low-, and high-level representation**   As the lower layer in all our generative models was linear (cf. Fig. 1), we could effectively study what each $\mathbf{z_1}$ dimension represents through their projective fields computed with a standard latent traversal procedure. In all of our Markovian generative models we have found a complete basis of localized oriented filters in $\mathbf{z_1}$. However, in SkipVAE, there is only an undercomplete basis present. For more details on the role of the sparse prior and on the analysis of the SkipVAE representation, see A.3.1 and A.3.2.

We studied the non-linear $\mathbf{z_2}$ representation in our models with the help of the texture dataset discussed in A.2.2. These texture images were constructed in such a way that the texture family cannot be linearly decoded from the pixels (see Fig. 2a, gray bar). However, it is a robust feature of all inspected hierarchical models that some $\mathbf{z_2}$ dimensions linearly encode high-level texture family information (see Fig. 2a). The texture-selective dimensions in $\mathbf{z_2}$ are actually special in the sense that by mildly compressing the representation with $\beta_2 = 1.25$ (as described in A.1.2) all other dimensions collapse (see Fig. 2b), indicating that these are dominant features of natural statistics.

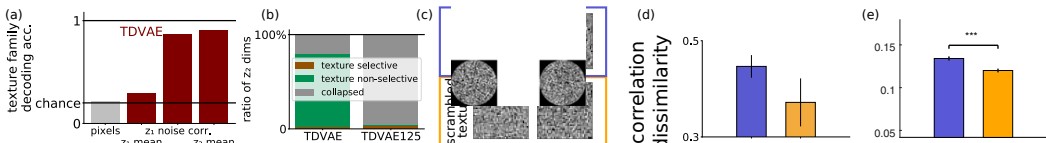

Figure 2: (a) Linear decodability of texture families from raw pixels, mean $\mathbf{z_1}$ and $\mathbf{z_2}$ activations, $\mathbf{z_1}$ noise correlations. (b) Collapse of texture non-selective dimension in a compressed $\mathbf{z_2}$ representation (TDVAE125). (c)-(e) Stimulus-statistics dependent noise correlations in TDVAE125 (d) and in macaques (e). (c) Top row: example of texture images. Bottom row: example of scrambled texture images (d) Noise correlation dissimilarities calculated between $\mathbf{z_1}$ units in TDVAE125. Noise correlation dissimilarities are calculated across five image pairs. See text for details. (e) Dissimilarity of pairwise noise correlations between responses for different stimuli within a particular condition (colors). Noise correlations are calculated from 80–120 trials using the same stimulus. Dissimilarities are averaged across image pairs. Reproduced with permission from [2], PNAS.

**Noise correlations**   Identifying the features represented in $\mathbf{z_2}$ provides an opportunity to investigate the contributions of top-down connections to $\mathbf{z_1}$ activations. Since $\mathbf{z_1}$ is in a linear generative relationship with images $\mathbf{x}$, it is not surprising that texture families are not decodable from $\mathrm{E}[q_\Phi(\mathbf{z_1} \,|\, \mathbf{x})]$ (see Fig. 2a)). Since $\mathbf{z_2}$ is assumed to contribute to shaping the posterior of $\mathbf{z_1}$, it is tempting to investigate if higher moments of $q_\Phi(\mathbf{z_1} \,|\, \mathbf{x})$ carry information about high-level features. By the construction of hVAEs, only those with top-down inference paths (TDVAE, SkipVAE) can represent such higher-order moments, while ChainVAE (featuring bottom-up inference) cannot. Indeed, the texture family can be linearly decoded from the posterior correlations of $\mathbf{z_1}$ in the TDVAE model (Fig. 2a):

$$\mathrm{corr}^{\mathrm{noise}}(\mathbf{x}) = \mathrm{corr}[q_\Phi(\mathbf{z_1} \,|\, \mathbf{x})] = \mathrm{corr}\left[ \int d\mathbf{z_2} q_\Phi(\mathbf{z_1} \,|\, \mathbf{x}, \mathbf{z_2}) \cdot q_\Phi(\mathbf{z_2} \,|\, \mathbf{x}) \right], \tag{2}$$

which we call noise correlations (NC), following the neuroscience literature terminology.

NC can be measured in population recordings of neuronal activity. As opposed to the traditional accounts that consider NC in the context of information theoretical arguments, we argue that these can be signatures of probabilistic computations: if neuronal population activity represents a posterior over the values of inferred features, then NC between pairs of neurons should display patterns similar to the patterns of correlations measured among the variables of the generative model. Therefore, we seek to identify NC in $\mathbf{z_1}$ and relate them to noise correlations in V1 population recordings.

Texture family decodability from NC suggests that image sets carrying high-level statistical information display stimulus-specific NCs. In contrast, an image set devoid of high-level information is expected to display less specificity in NCs. Assuming that hierarchical inference shapes the response distribution in the visual cortex, this prediction implies modulations in the stimulus specificity of NCs in neuronal recordings. To do this, we created stimuli from texture images such that the high-level structure was removed by permuting the mean activations of the $\mathbf{z_1}$ variational posterior (filter scrambling). NC specificity was characterized by the dissimilarity of NC matrices (L1 norm of NC matrix differences) for pairs of images drawn from texture images (top panel in Fig. 2c) and also from filter scrambled images (bottom panel in Fig. 2c). As predicted, NC dissimilarity was reduced with filter scrambling (Fig. 2d). This prediction is consistent with NC measurements [2] in a population of V1 neurons when macaques attended texture and filter scrambled images (Fig. 2e).

**Image inpainting and illusory contours**   We demonstrate here that top-down effects enhance the robustness of the low-level representation $\mathbf{z_1}$ to out-of-distribution stimuli. First, we attempted image inpainting of masked texture images with TDVAE, and also with ChainVAE and LinearVAE to see the relevance of top-down architecture and multiple layers. However, note that image inpainting is just a downstream task for our models, unlike for state-of-the-art supervised image completion models (such as [18]). We inferred $\mathbf{z_1}$ from the masked textures and generated inpainted reconstructions. To test how much contextual information is present in the pixels inside the mask of the inpainted image, we tested how well the texture family can be decoded from them (see Fig. 3 for the result with various mask radii). We can see that the decoding accuracy is well above chance (0.2) for all models but is higher for TDVAE than for models without top-down connection or with a single layer. We observed the same for natural images where an alternative evaluation metric was used (see in A.4).

Instead of seeking neurophysiological experiments using masked natural image stimuli, we studied our models' responses to illusory contour stimuli inspired by [28]. In that work, Kanizsa square images (second row in the left panel of Fig. 4a) were shown to macaques and the V1 activity of the primates was measured. It was shown that particular V1 cells that are responsive to real edges were also firing to the illusory edge. We demonstrate here the relevance of top-down effects in this phenomenon by comparing the mean responses of TDVAE and ChainVAE/LinearVAE (models without top-down connection or with a single layer only) to an illusory edge.

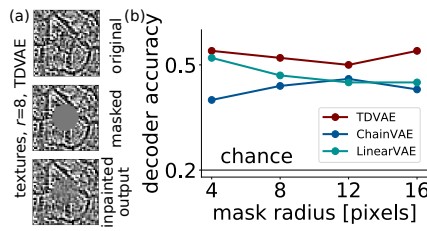

Figure 3: (a) Setup of masking experiment. (b) Linear decodability of texture families from inpainted reconstructions.

We designed cropped Kanizsa square stimuli by aligning the illusory contour to $z_1$ filters, as well as "rotated corners" stimuli as control stimuli (Fig. 4(b), see A.5 for details of filter selection). Note that, in contrast to the measurements in [28], we used only the lower half of the Kanizsa square stimuli. The fitted stimuli were then moved perpendicular to the filter orientation and the per-filter posterior mean $z_1$ responses were recorded as a function of the stimulus position (Fig. 4b). The response curves were averaged for the selected filters and were plotted for different models and stimuli (Figs. 4c–e) (the analogous figure is reproduced with permission from [28](PNAS) in Fig. 4a).

We found that 1) the mean responses to the illusory contour stimuli were substantially larger than those to the "rotated corners" stimuli (as in [28]), and 2) the mean responses to the illusory contour stimuli relative to that to the line stimuli was substantially larger in TDVAE than in ChainVAE/LinearVAE (Figs. 4c–e). Thus, the response of ChainVAE does not exceed the intensity of linear responses, but the presence of top-down inference significantly enhances the responses to the illusory contour.

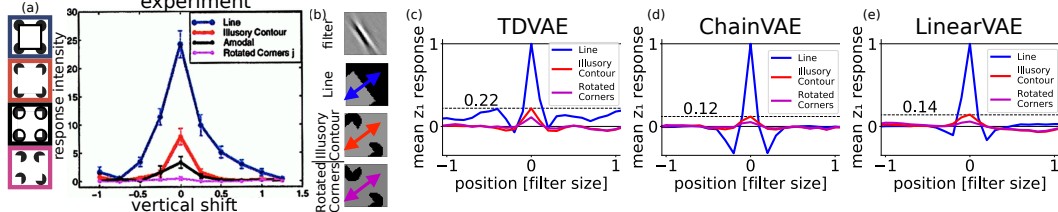

Figure 4: (a) Intensity of V1 cells in macaques as a response of real and illusory contour stimuli as a function of vertical position of the edge (reproduced with permission from [28], PNAS, Copyright (2001) National Academy of Sciences, U.S.A.). (b) Illusory contour, line and rotated corners stimuli fitted automatically to a selected filter. The arrows denote the direction of gradual displacement of the stimuli during the experiment. (c–e) Mean activations of $z_1$ in response to line, illusory contour and rotated corners stimuli stimuli in TDVAE, ChainVAE, and LinearVAE.

## 4  Conclusions

We modeled the probabilistic representation of the early visual system V1/V2 by learning a hierarchical VAE on natural images. We saw that apart from choosing a sparse prior for the lower layer, we had to choose a VAE with a purely Markovian generative model (unlike the state-of-the-art deep VAE architectures) to get a complete basis of Gabor filters. In addition, we showed that texture-like features emerge that are robust against compression or architectural change. We studied the stimulus sensitive noise correlations of the lower layer. We showed that in models featuring a top-down recognition model, unlike the mean of the posterior, NC contains high-level information: texture family information was linearly decodable from them. We also demonstrated that, in line with V1 measurements in macaques [2], stimulus specificity of NC decreases once high-level information is removed from the images. We demonstrated the role of top-down inference in robustness of the representation against distortions. In both the texture inpainting and the illusory contour detection experiment, TDVAE performed the best. The latter experiment had findings similar to measurements using Kanizsa stimuli in macaques [28].

Predictive coding (PC) has been proposed as an alternative form of generative model for the visual cortical hierarchy that features top-down connections [31, 5, 29]. Similarly to our case, these models have also been proposed to account for illusory contours [29, 34]. Our model differs from PC in two fundamental points: 1) Top-down contributions have a purely computational-level motivation, without specific algorithmic-level assumptions; 2) Our framework extends the scope of PC to probabilistic computations, and thus can account for patterns in noise correlations.

We believe that some of our findings (inspired by neuroscience domain knowledge) can also be valuable to the machine learning community. Representation of (almost exclusively single-layer) VAEs is an area actively studied. However, the higher moments of the posterior are ignored. We demonstrated that correlation between latents can contain interesting information. We also showed that an hVAE with a Markovian generative model learns a more interpretable representation compared to non-Markovian hVAEs (which achieve state-of-the-art image generation quality), where feature hierarchy is less pronounced. Finally, we highlighted the relevance of the top-down recognition model in image inpainting. In future work, we seek to explore the response properties of higher visual areas using larger image patches and deeper Markovian generative models.

## Acknowledgments and Disclosure of Funding

We are grateful to Mihály Bányai and Dávid G. Nagy for helpful discussions. We would also like to thank the reviewers for their constructive feedback. Funding in direct support of this work: Human Frontier Science Program (RGP0044/2018), and the Artificial Intelligence National Laboratory of Hungary (NKFIH-1530-4/2021).

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

# A Appendix

## A.1 Model details

### A.1.1 Factorization of the variational posterior

For a hierarchical generative model with two layers, one can factorize the true posterior in two ways. In the bottom-up fashion this reads:

$$p(\mathbf{z_1}, \mathbf{z_2} \,|\, \mathbf{x}) = p(\mathbf{z_2} \,|\, \mathbf{x}, \mathbf{z_1}) \cdot p(\mathbf{z_1} \,|\, \mathbf{x}). \tag{3}$$

For the Markovian case, this simplifies to a chain, as $p(\mathbf{z_2} \,|\, \mathbf{x}, \mathbf{z_1})$ is independent of $x$. For both the Markovian and non-Markovian cases, the posterior can be factorized in a top-down manner as well:

$$p(\mathbf{z_1}, \mathbf{z_2} \,|\, \mathbf{x}) = p(\mathbf{z_1} \,|\, \mathbf{x}, \mathbf{z_2}) \cdot p(\mathbf{z_2} \,|\, \mathbf{x}). \tag{4}$$

We choose to have a factorization of the variational posterior which mimic one of the above forms of factorization of the true posterior.

Note that it is not compulsory to choose a factorization of the variational posterior which preserves Eq. (3) or (4). It would mean though that we cannot saturate the ELBO even if we have a very expressive distribution for the single-layer variational posteriors. For example, in [39] the authors choose to work with $q_\Phi(\mathbf{z_1}, \mathbf{z_2} \mid \mathbf{x}) = q_\Phi(\mathbf{z_1} \mid \mathbf{x}) \cdot q_\Phi(\mathbf{z_2} \mid \mathbf{x})$. For them, the hierarchical nature of the representation comes from choosing a more compressing network for $q_\Phi(\mathbf{z_2} \mid \mathbf{x})$ compared to $q_\Phi(\mathbf{z_1} \mid \mathbf{x})$.

In the top-down case, we will have a simple functional form (for example, diagonal normal or Laplace distribution) for $q_\Phi(\mathbf{z_2} \mid \mathbf{x})$ and $q_\Phi(\mathbf{z_1} \mid \mathbf{x}, \mathbf{z_2})$ and optimize for the TD objective function (for the Markovian generative model):

$$
\begin{aligned}
\mathcal{F}_{\mathrm{TD}}(\mathbf{x}, \theta, \Phi) \;=\; & \mathrm{E}_{q_\Phi(\mathbf{z_2} \mid \mathbf{x}) q_\Phi(\mathbf{z_1} \mid \mathbf{x}, \mathbf{z_2})}[\log p_\theta(\mathbf{x} \mid \mathbf{z_1})] - \\
& - \; \beta_1 \cdot \mathrm{E}_{q_\Phi(\mathbf{z_2} \mid \mathbf{x})}[\mathrm{KL}[q_\Phi(\mathbf{z_1} \mid \mathbf{x}, \mathbf{z_2}) \,\|\, p_\theta(\mathbf{z_1} \mid \mathbf{z_2})]] - \\
& - \; \beta_2 \cdot \mathrm{KL}[q_\Phi(\mathbf{z_2} \mid \mathbf{x}) \,\|\, p_\theta(\mathbf{z_2})].
\end{aligned}
\tag{5}
$$

For the non-Markovian case, the only difference is that in the reconstruction term $p_\theta(\mathbf{x} \mid \mathbf{z_1})$ needs to be replaced by $p_\theta(\mathbf{x} \mid \mathbf{z_1}, \mathbf{z_2})$. Note that the KL term is a sum of layer-wise KL terms. Recent deep hVAEs use this latent posterior structure (with a non-Markovian generative model) [41, 9].

We introduced the parameters $\beta_1$ and $\beta_2$. When $\beta_1 = \beta_2 = 1$, the objective function is identical to the ELBO. See the role for $\beta$-s different from 1 in the next subsection.

If we choose bottom-up factorization we constrain $q_\Phi(\mathbf{z_1} \mid \mathbf{x})$ to be of a simple form and the ELBO becomes for the Markovian case (since we only consider this type of generative model with the bottom-up recognition model):

$$
\begin{aligned}
\mathcal{F}_{\mathrm{BU}}(\mathbf{x}, \theta, \Phi) \;=\; & \mathrm{E}_{q_\Phi(\mathbf{z_1} \mid \mathbf{x})}[p_\theta(\mathbf{x} \mid \mathbf{z_1}) + \beta_1 \cdot p_\theta(\mathbf{z_1} \mid \mathbf{z_2})] - \beta_1 \cdot \mathrm{H}[q_\Phi(\mathbf{z_1} \mid \mathbf{x})] \\
& - \; \beta_2 \cdot \mathrm{E}_{q_\Phi(\mathbf{z_1} \mid \mathbf{x})} \mathrm{KL}[q_\Phi(\mathbf{z_2} \mid \mathbf{z_1}) \,\|\, p_\theta(\mathbf{z_2})],
\end{aligned}
\tag{6}
$$

where H denotes the entropy of the distribution.

### A.1.2 The role of $\beta$-s

We allowed the KL-type terms to be scaled in Eq. (5) and (6) with parameters $\beta_1$ and $\beta_2$. When $\beta_1 = \beta_2 = 1$ the objective function is identical to the ELBO. One reason for doing this is practical. It can help the training process by allowing these parameters to slowly increase from a small value to 1 ($\beta$ annealing). Furthermore, from a representation learning point of view, $\beta$-s can shape the latent representation by manipulating the mutual information between the observed and the latent variables [7, 1].

Single-layer $\beta$-VAEs [7] (where $\beta$ is different from 1) has been extensively explored to see how it can contribute to the emergence of a more disentangled representation. This manipulation was shown to correspond to altering the capacity of the latent representation when interpreting the inference as lossy compression [1]. Our investigations concern the $\beta_2 > 1$ case, since it establishes an inductive bias to learn a compressed representation at the higher latent layer.

### A.1.3 Architectural details

The computational graph of the recognition model for the TDVAE and SkipVAE are non-trivial and contains further inductive biases (Fig. 5). We can see that there is a direct connection from $\mathbf{x}$ to the stochastic variable $\mathbf{z_2}$ in the sense that the latter does not depend on $\mathbf{z_1}$. We emphasize that this property of the recognition model is compatible with both Markovian and non-Markovian generative models.

First, we discuss the details of the architectural choices for the top-down recognition model present in TDVAE and SkipVAE. There are four MLPs defined for the recognition models as depicted in Fig. 5. The first neural network (*MLP.a*) maps the pixel space to a layer $L_x$ that is shared between the computations of $q_\Phi(\mathbf{z_2} \mid \mathbf{x})$ and $q_\Phi(\mathbf{z_1} \mid \mathbf{x}, \mathbf{z_2})$. From $L_x$ *MLP.b* computes the mean and variances

Table 1: Models presented in this paper.

| Name | Architecture | Patch size | $\mathbf{z_1}$ distr.'s | dim($\mathbf{z_1}$) | dim($\mathbf{z_2}$) | $\beta_1$ | $\beta_2$ |
|---|---|---|---|---|---|---|---|
| LinearVAE | LinearVAE | $40 \times 40$ | Laplace | 1800 | N/A | 1.00 | N/A |
| TDVAE | TDVAE | $40 \times 40$ | Laplace | 1800 | 250 | 1.00 | 1.00 |
| TDVAEn | TDVAE | $40 \times 40$ | normal | 1800 | 250 | 1.00 | 1.00 |
| TDVAE125 | TDVAE | $40 \times 40$ | Laplace | 1800 | 250 | 1.00 | 1.25 |
| SkipVAE | SkipVAE | $40 \times 40$ | Laplace | 1800 | 250 | 1.00 | 1.00 |
| ChainVAE | ChainVAE | $40 \times 40$ | Laplace | 1800 | 250 | 1.00 | 1.00 |

Table 2: Number of hidden units in each MLP layer computing the mean and the standard deviation of each conditional distribution in the bottom-up recognition models.

| Name | Architecture | $p_\theta(\mathbf{z_1} \mid \mathbf{z_2})$ | $q_\Phi(\mathbf{z_1} \mid \mathbf{x})$ | $q_\Phi(\mathbf{z_2} \mid \mathbf{z_1})$ |
|---|---|---|---|---|
| LinearVAE | LinearVAE | N/A | (2000, 2000) | N/A |
| ChainVAE | ChainVAE | (2000) | (2000, 2000) | (1000, 500, 250) |

of the $q_\Phi(\mathbf{z_2} \mid \mathbf{x})$ distribution. The third MLP (*MLP.c*) transforms $\mathbf{z_2}$ to layer $L_z$. We fuse the information from $\mathbf{x}$ and $\mathbf{z_2}$ by concatenating $L_x$ and $L_z$ and apply an MLP on the combined layer to obtain the mean and variances of $q_\Phi(\mathbf{z_1} \mid \mathbf{x}, \mathbf{z_2})$ (*MLP.d*). The number of hidden layers and hidden units used in each MLP to calculate the means and standard deviations of the conditional generative and variational posterior distributions is shown in Table 2 for models with bottom-up recognition models and in Table 3 for models with top-down recognition models.

We also tested the significance of parameter sharing in the encoder *MLP.a*. In a control experiment, we turned off parameter sharing in $q_\Phi(\mathbf{z_2} \mid \mathbf{x})$ and $q_\Phi(\mathbf{z_1} \mid \mathbf{x}, \mathbf{z_2})$ and found that the ELBO, the dimensions of the learned representation, and the decoding accuracies changed by less than 2%. This means that parameter sharing through the shared encoder *MLP.a* has only a marginal effect on the learned model.

### A.1.4 Model training details

We trained several model instances representing the model architectures in Section 2 on whitened natural image patches. The main hyperparameters of a selected subset of experiments is shown in Table 1. We trained our models with the Adam optimizer [25]. We found that while the learned $\mathbf{z_1}$ representation was robust against regularization techniques (we tested weight decay, gradient clipping and gradient skipping), the learned $\mathbf{z_2}$ representation was sensitive to these. To eliminate such regularization artifacts, we turned off

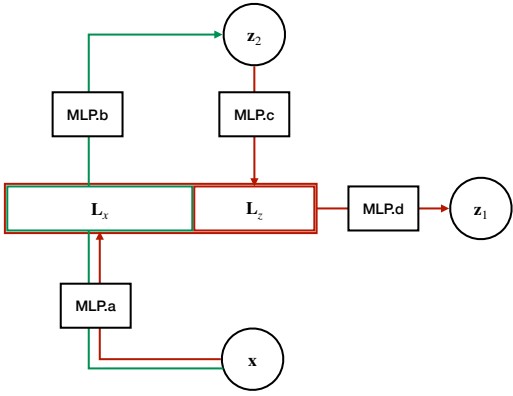

Figure 5: Detailed illustration of the recognition model for TDVAE and SkipVAE. The green/red arrows highlight the computational flow for calculating the parameters of the distributions $q_\Phi(\mathbf{z_2} \mid \mathbf{x}) / q_\Phi(\mathbf{z_1} \mid \mathbf{x}, \mathbf{z_2})$. The weights of MLP.a that produce the intermediate layer $L_x$ are shared between the two computations. The intermediate layers $L_x$ and $L_z$ are concatenated to form the input to MLP.d.

Table 3: Number of hidden units in each MLP layer computing the mean and the standard deviation of each conditional distribution in the top-down recognition models.

| Name | $p_\theta(\mathbf{z_1} \mid \mathbf{z_2})$ | MLP.a | MLP.b | MLP.c | MLP.d | skip |
|---|---|---|---|---|---|---|
| TDVAE | (2000) | (2000) | (1000, 500, 250) | (250, 500, 1000, 2000) | (2000) | N/A |
| TDVAEn | (2000) | (2000) | (1000, 500, 250) | (250, 500, 1000, 2000) | (2000) | N/A |
| TDVAE125 | (2000) | (2000) | (1000, 500, 250) | (250, 500, 1000, 2000) | (2000) | N/A |
| SkipVAE | (2000) | (2000) | (1000, 500, 250) | (250, 500, 1000, 2000) | (2000) | (2000, 1800) |

Table 4: Model optimization parameters.

| Name | Architecture | Learning rate(s) |
|------|-------------|------------------|
| LinearVAE | LinearVAE | $3 \times 10^{-5}$ |
| TDVAE | TDVAE | $5 \times 10^{-5} \to 2.5 \times 10^{-5}$ |
| TDVAEn | TDVAE | $5 \times 10^{-5}$ |
| TDVAE125 | TDVAE | $5 \times 10^{-5}$ |
| SkipVAE | SkipVAE | $2 \times 10^{-5}$ |
| ChainVAE | ChainVAE | $1 \times 10^{-5}$ |

Table 5: Source for the texture family seed images and preprocessing parameters. For details, see text.

| Texture family | Origin | Filename | Channel | Subsampling |
|----------------|--------|----------|---------|-------------|
| 0 | textures.com | FoodGrains0001_1_seamless_S | green | $4 \times 4$ |
| 1 | textures.com | Leather0028_1_M | green | $2 \times 2$ |
| 2 | textures.com | SoilCracked0079_1_seamless_S | red | $2 \times 2$ |
| 3 | textures.com | Carpet0025_1_seamless_S | blue | $2 \times 2$ |
| 4 | [6] | D111 | N/A | N/A |

weight decay and increased gradient clipping and skipping thresholds to have an activation frequency below $10^{-6}$. As a final step, we continued to decrease the learning rate (constant in each experiment) until the learned representation in $\mathbf{z_2}$ converged. The test ELBOs were at most 20% higher than the training ELBOs, demonstrating that the number of training examples ($3.2 \times 10^5$) was enough to avoid overfitting. Training the set of models in Table 1 took 322 hours altogether on a computer equipped with one Nvidia GeForce RTX 3080 Ti GPU. The source code of our models contains code from [36] which uses the Apache-2.0 license.

## A.2  Datasets

### A.2.1  Natural image data

We sampled $3.2 \times 10^5$ training and $6.4 \times 10^4$ test images from the van Hateren database [42], matched their grand total intensity histograms to the unit normal distribution, and applied the whitening procedure described in [3]. This whitening procedure discards $100(1 - \pi/4)\%$ of the high-frequency PCA components, keeping 1256 data dimensions in $40 \times 40$ image patches (see Fig. 6a for examples).

### A.2.2  Texture data

Natural images are known to feature a characteristic linear structure [32, 38]. Higher-order dependencies that cannot be captured by the linear model [43, 35] and nonlinear features have also been identified in natural images [38, 24]. This ensures that these provide an exquisite test bed for investigating emerging representations in hierarchical generative models.

Synthetic data sets were generated by an optimization algorithm developed by Portilla and Simoncelli[35] and whitened with the same procedure as the natural data sets (see Fig. 6b for examples). The seed images of the five texture families were downloaded from `https://www.textures.com/` under a license that granted free usage for noncommercial purposes, as well as from [6].

Table 5 shows the source and preprocessing parameters for the seed images used to synthesize the texture families. Colored seed images (0) — (3) were turned into grayscale by selecting one of their color channels, then cubic subsampling was applied with a window size that ensured that the dominant Fourier component fits into a $40 \times 40$ cropped image patch. The grayscale seed image (4)

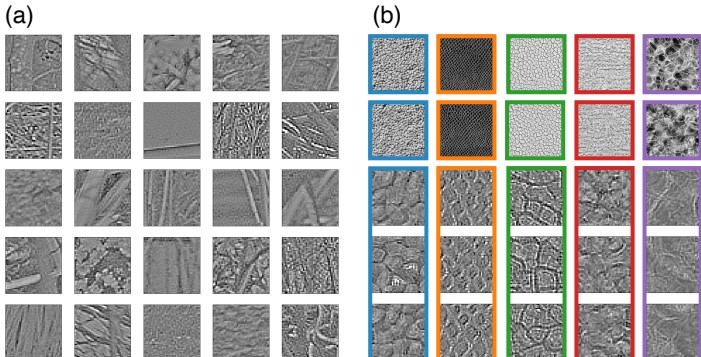

Figure 6: (a) Example $40 \times 40$ whitened natural image patches, with matching grey scale. (b) First row: $256 \times 256$ crops from the five texture family seed images after preprocessing with the parameters in Table 5. Second row: $256 \times 256$ crops from textures synthesized with [35]. Bottom rows: Examples of $40 \times 40$ whitened texture patches cropped from synthesized texture images.

Table 6: Parameters and download URLs for the training and test datasets used in the paper. See text for details.

| Image type | Patch size | Link |
|---|---|---|
| natural | $20 \times 20$ | download |
| natural | $40 \times 40$ | download |
| textures | $20 \times 20$ | download |
| textures | $40 \times 40$ | download |

has no characteristic Fourier component (it has a scale-free autocorrelation function); therefore, no subsampling was applied. The seed images preprocessed in this way are shown in the top row of Fig. 6(b).

The preprocessed texture seed images were then fed into the texture synthesis method [35] one by one to generate a large number of texture images for each texture family (for samples, see the second row of Fig. 6(b)). These synthesized texture images were then used to generate a large number of $40 \times 40$ pixel whitened texture patches (similarly to the natural image data set). To promote statistical correspondence to our natural training images, we only used texture families on which a sparse LinearVAE model (see Tab. 1) learned a complete basis of localized, oriented, bandpass filters.

The resulting datasets were placed into a public repository that ensures long-term preservation of the data, provides a Digital Object Identifier, and publishes metadata in several metadata standards, including Schema.org and DCAT. In the interest of anonymity, we provide only anonymized links in Table 6. Each file is in pickle format and was generated with Python 3.8.5. Each file contains a Python dictionary with the following fields:

**'train_images'** 640,000 float32 images used for model training. $20 \times 20$ pixel images contain 400 pixel intensities, and $40 \times 40$ pixel images contain 1600 pixel intensities each.

**'train_labels'** float32 labels for each image in **'train_images'**. All natural images are labeled with 0.0. Texture images are labeled with 0.0, 1,0, 2.0, 3.0, or 4.0, according to their texture family.

**'test_images'** 64,000 float32 images used for model testing. $20 \times 20$ pixel images contain 400 pixel intensities, $40 \times 40$ pixel images contain 1600 pixel intensities each.

**'test_labels'** float32 labels for each image in **'test_images'**. All natural images are labeled with 0.0. Texture images are labeled with 0.0, 1,0, 2.0, 3.0, or 4.0, according to their texture family.

These data are published under the terms of the Creative Commons Attribution 4.0 International license.

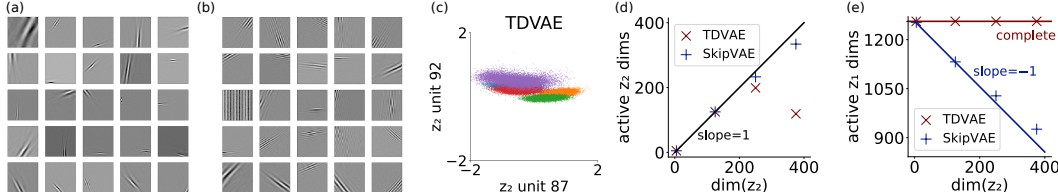

Figure 7: (a) $z_1$ projective fields using sparse (Laplace) priors in latent model layer $z_1$. (b) Same with normal prior. A sparse $z_1$ prior leads to more localized $z_1$ projective fields than a normal $z_1$ prior, while the sparsity of the $z_2$ prior does not have a noticeable effect. (c) A subset of dimensions of $E[q_\Phi(z_2 \mid x)]$ are texture family selective in all studied two-layer models. Different colors represent responses to images from different texture families (d) Active $z_2$ dimensions increase with increasing $\dim(z_2)$ in SkipVAE but not in TDVAE. (e) TDVAE learns complete linear bases in $z_1$, irrespective of $\dim(z_2)$. SkipVAE moves as many of these to $z_2$ as possible.

## A.3 Details on the hierarchical representation

### A.3.1 Low-level layer

In each model we found that the $z_1$ dimensions were clearly clustered into two groups (active and collapsed) based on the mean squared intensity of their projective fields. In image reconstruction experiments, collapsed $z_1$ dimensions were responsible for less than $10^{-6}$ of the pixel variances generated, that is, their contribution was negligible. The number of active $z_1$ dimensions in all Markovian models (LinearVAE, TDVAE, and ChainVAE) was equal to the data dimensions (1256). Therefore, the active $z_1$ components form a complete linear basis in the space of the training images. In contrast, the active $z_1$ dimensions in the non-Markovian SkipVAE models always formed an undercomplete basis only.

Training SkipVAE models with different numbers of $z_2$ dimensions, we found that the sum of active latent dimensions is constant (see Fig. 7(d)-(e)). This reveals a fundamental difference between the representations learned by models with Markovian and non-Markovian generative models. In the Makovian case, the generative models force all low-level linear features into $z_1$, cleanly separating them from nonlinear, possibly higher-level features in $z_2$. However, in the non-Markovian model, the complete low-level representation is distributed between $z_1$ and $z_2$ due to the generative skip connection. This reduces both the interpretability of the non-Markovian model compared to the Markovian ones and the efficiency of inductive biases in shaping the representations.

We found that the qualitative character of the projective fields of active $z_1$ dimensions depends on the probability distributions chosen for the $z_1$ generative and recognition models. The sparse Laplace distribution leads to localized, oriented, bandpass, Gabor-like filters with low uncertainties (Gabor wavelets are commonly defined by having the lowest possible uncertainty value). They are reminiscent of the Gabor-like filters found in single-layer sparse linear models of natural images [32, 3] (see Fig. 7(a)). Using normal distribution results in more extended oriented filters, akin to the Fourier PCA components of natural images (see Fig. 7(b)). However, we found that the representation of $z_1$ was not affected by the choice of $z_2$ prior (Laplace or Normal). We found one exception to this rule: a minority of $z_1$ dimensions in TDVAE display nonlocal projective fields with high uncertainties, which are accompanied by a large number of texture-non-selective active dimensions in $z_2$. With a slight increase of $\beta_2$, all $z_1$ filters become Gabor-like and all active $z_2$ dimensions become texture-selective (see Fig. 2(b)).

### A.3.2 High-level layer

The $z_2$ dimensions clustered into an active and a collapsed group, the latter characterized by small variances of the posterior $z_2$ means and close to unit means of the posterior $z_2$ variances within each texture family. Training models with different numbers of $z_2$ dimensions revealed that there is a limit to the number of active $z_2$ dimensions in TDVAE models. SkipVAE models, however, use all available $z_2$ dimensions (Fig. 7d). This is because low-level filters appear in $z_2$, as discussed in the previous subsection. We have seen that there are texture-sensitive $z_2$ dimensions. See (Fig. 7c),

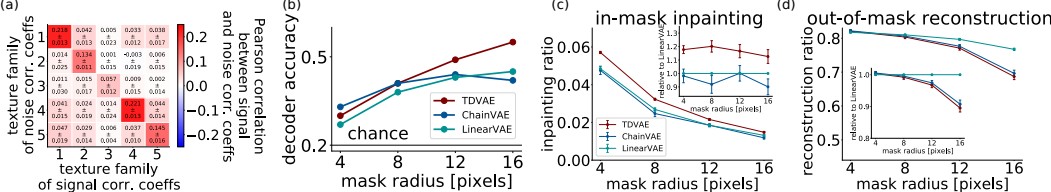

Figure 8: (a) Correlations between $\mathbf{z_1}$ signal and noise correlation coefficients are significantly stronger within than across texture families. (b) Linear decodability of texture families from inpainted reconstructions without cropping the 40px image to 20px. (c) Top-down inference substantially enhances in-mask image completion. (d) Top-down inference somewhat degrades out-of-mask reconstruction.

where the clustering of texture families is shown in two dimensions. The texture sensitivity at the level of $\mathbf{z_2}$ was persistent in the two choices of the $\mathbf{z_2}$ prior. The number of texture-sensitive units was slightly lower for a sparse prior. These units tended to rotate toward the $\mathbf{z_2}$ axes, resulting in a more sparse and disentangled representation.

### A.3.3 Signal vs. noise correlations

Since the model parameters were learned from training data, it is plausible that the uncertainties expressed in the posteriors of $\mathbf{z_1}$ should reflect the properties of the stimulus. [24] highlighted that texture families possess characteristic correlations in the means of linear filter activations

$$\text{corr}^{\text{signal}}(\text{tf}) = \text{corr}_{\text{p}_{\text{tf}}(\mathbf{x})}[\text{E}[q_\Phi(\mathbf{z_1} \mid \mathbf{x})]], \tag{7}$$

termed signal correlations ($\text{tf}$ denotes texture family). This is also characteristic of all of our models. We found that within each texture family, the elements of the signal correlation and the texture-averaged noise correlation matrices (corresponding to filter pairs) are positively correlated. This correlation is significantly stronger within than across texture families (Fig. 8a).

### A.4 Metrics for image inpainting performance

**Texture family decodability** In Section 3 and Fig.3 we characterized the image inpainting performance on textures with texture family decodability. Here, we describe in detail how this accuracy was obtained. To focus on the region inside the mask, we took the inpainted image and zeroed the pixel values *outside* of the central disk used for masking. We then cropped the central $20 \times 20$ pixel part of the image. This cropped image was fed to a TDVAE20 model, which was previously trained on 20px natural images. The posterior mean values of $\mathbf{z_2}$ were calculated and a logistic regression model was trained on them to decode texture family information.

For a larger mask radius, image inpainting is harder. However, more pixels are available for the decoder to decode the texture family. This is why there is no obvious trend in the accuracy values as a function of the mask radius in Fig.3.

We also checked the above metric without cropping a 20px piece from the 40px inpainted image and using a 40px TDVAE for calculating $\mathbf{z_2}$ posteriors. Fig. 8b shows that TDVAE performs better in inpainting compared to models without top-down connections. Without cropping, this effect becomes noticeable when the mask radius is large enough since the masked area is larger and the inpainting task is harder.

**Regression based metric** We also performed the image inpainting experiment on natural images. In that case, we cannot use a classification-based performance metric. Therefore, we took the function of the pixel values of the inpainted image against the pixel values of the original unmasked image. Then, two separate lines were fitted to this function restricted to either inside or outside the mask area. The slope of the inside-mask line is an indication of inpainting quality, and we can see in Fig. 8c that two-layer top-down hVAEs perform better compared to a single-layer LinearVAE and two-layer hVAE with bottom-up recognition model. However, it could be that this is not purely due to the top-down effect, but these models are simply better at reconstructing any images. To control for that,

we also checked the slopes of the outside-mask line. We can see in Fig. 8d that outside the mask the LinearVAE reconstruction is better compared to TDVAE. This means that the top-down models rely more on higher-level features when filling in the mask.

## A.5   Filter selection for illusory contour experiment

For each model studied, we performed the following steps. 1) We automatically selected all $z_1$ filters fit for measuring their response to illusory contour, line and "rotated corners" stimuli. We used the following filter selection criteria: central position, small or medium size, and medium wavelength. Central position, small or medium size, and not too large wavelength were needed so that the stimuli fit into the area of the image patch. Small (just a few pixel) wavelengths were excluded to avoid pixel aliasing effects. The illusory contour stimuli were constructed to have their potential illusory contours in the exact positions of the real contours of the corresponding line stimuli. 2) The orientation, initial position, and size of each illusory contour, line and "rotated-corners" stimulus was fitted individually to the orientation, center, and size of each selected filter, respectively (see an example in Fig. 4b). 3) The fitted stimuli were moved perpendicular to the filter orientation, and the posterior mean $z_1$ responses per filter were recorded as a function of the stimulus position. 4) Response curves were averaged for the selected filters and plotted for different models and stimuli (Figs. 4c–e).

