# OpenReview forum: "Top-down effects in an early visual cortex inspired hierarchical Variational Autoencoder"
_NeurIPS.cc/2022/Workshop/SVRHM — SVRHM Oral_

### Official Review · Reviewer_xJ6A · 2022-10-08
**Review: Study of top-down visual effects with hierarchical VAEs**

**Rating:** 9
**Confidence:** 3

**Review:**

**Summary**: The authors perform an extensive study of multiple choices of hierarchical VAE architectures inspired by the recent literature, and find a proposed method (TDVAE) which matches many observed properties of biological visual systems.

**Positives**: This work is a good comparison of multiple hVAE frameworks and specifically an in-depth investigation of which representations they learn. This type of study is somewhat lacking in the literature compared to other generative modeling frameworks which have also been compared with the early visual system (such as ICA). The detailed appendix is very much appreciated for this kind of empirical study as well. Finally, I find the illusory contour experiment to be the most interesting and novel, and commend the authors for devising it.

**Areas for improvement**: The section on noise correlations was a bit dense and challenging to understand for someone not familiar with the relevant neuroscience literature.

**Questions**: Can the authors describe why they measured image accuracy in the image in-painting experiment rather than using a metric like likelihood? Perhaps I am misunderstanding something about the experiment setup.

**Conclusions**: Overall I think this is a very relevant paper which attempts to bridge modern generative models used in machine learning with many observations from neuroscience. I strongly recommend acceptance.

---

> ### Author Response · Authors · 2022-12-23
> **response to reviewer xJ6A**
>
> We would like to thank the reviewer for the feedback. We have extended the section on noise correlation for improving clarity in the camera-ready version of the manuscript.
> In the in-painting experiment, we used texture-decoding accuracy because we were interested in how semantic the reconstruction is (not necessarily how the pixel-by-pixel L2 distance between the reconstruction and the original image).

---

### Official Review · Reviewer_LBTN · 2022-10-14
**The goal of this work is to investigate the properties and representations of the early visual cortex. To tackle this problem a two layer hierarchical VAE was proposed whose generative part is Markovian. The paper reads well and the bio inspired approach seems relevant to the scope of the workshop.**

**Rating:** 6
**Confidence:** 2

**Review:**

The goal of this work is to investigate the properties and representations of the early visual cortex. To tackle this problem a two layer hierarchical VAE was proposed whose generative part is Markovian. The motivation of choosing a Markovian generative model is inspired from neuroscience which encourages interpret-ability in representation learning and establishes a hierarchy between latent layers. To quantify the efficacy of the proposed method, experiments were performed on natural images with applications such as noise correlation for low level representation learning and image in painting and illusory contours for high level representation learning. Results indicated the TDVAE perform better than Linear, Chain and Skip VAEs.




Pros.

1. The problem statement is relevant and interesting for the community. It is indeed important to study and analyze the representations of VAEs in order to better understand and establish links between generative modelling and neuroscience.

2. The proposed method is biologically inspired and the performed experiments (although preliminary) seem to back the claims made to analyze and link the latent representations in generative modeling with the understandings of the visual cortex from neuroscience.

3. The other baselines for comparison such as Linear VAE, Skip VAE and Chain VAE are technically correct from the proof of idea point of view.

Cons:

There are some terms used in the paper which I feel can be explained in a better way.

1. Lines 19, 20 " Generative models of natural images to understand....." What does Generative model of natural images mean?
2.  If the design choices mentioned in the paper for each VAE can be explained that would have been better. For each application mentioned how will what kind of design choices will be optimal? I am specifically asking about neural network architecture in terms of types of layers, number of layers, depth, filters, etc.

3. In order to scale to more meaningful image datasets for the same set of applications, how would the authors decide to approach?

4. For more clarity of the work, comparisons in terms of experiments with Predictive Coding is highly encouraged.

---

> ### Author Response · Authors · 2022-12-23
> **response for reviewer LBTN**
>
> We would like to thank the reviewer for their helpful comments and questions. Reflecting on the questions in the “Cons” section:
> 1. We mean a generative model (of any kind) which generates natural images. Or in other words, a generative model which is trained on natural images.
> 2. We extensively studied the performance of models with different stochastic variable structures (i.e. LinearVAE, SkipVAE, TDVAE, ChainVAE), although not all of our results appeared in this manuscript due to the length constraint. We found that
> a) Markovian structure (ChainVAE or TDVAE) is needed in order to find the complete basis of Gábor filters in z1
> b) Texture selective representation in z2 appeared in all of our two-layer models
> c) Noise correlations of z1 are only non-zero in models with top-down inference.
> d) Top-down inference models performed better on the masking experiments (both TDVAE and SkipVAE)
> However, we agree with the reviewer that the effect of architectural choices (like the depth/width/layer types of each part in the computational graph) could be studied more extensively on each application in the future.
> 3. To scale to significantly higher image resolution we will need to introduce more stochastic layers which would correspond to higher cortical areas. We used dense neural network layers in our experiments in order to introduce any inductive bias. For scaling, we will need to find a suitable convolutional architecture.
> Also, in our two-layer models, we see an important impact on whether the generative model was Markovian or not (complete basis in z1 only appeared in the case of a Markovian model). However, deep hierarchical VAEs currently have non-Markovian structure.
> 4. We fully agree with this.

---

### Official Review · Reviewer_rFXb · 2022-10-14
**Very interesting findings by comparing representations learned by hierarchical VAEs and the visual system**

**Rating:** 9
**Confidence:** 4

**Review:**

In this very interesting study, the authors compare representations learned by a collection of hierarchical VAE models with the visual system.

Contributions (- neutral, + strong, ++ very strong):
- (-) like classical deep nets, the upper layer of their hierarchical VAE models learns relevant high-level features, useful for example for texture detection.
- (+) unlike deep nets, but like alternative predictive coding models, their models can also account for inpainting capabilities of the visual system, and illusory edge representations in V1.
- (++) unlike other models, their models can also account for known properties of noise correlations in the early visual system, interpreted here as probabilistic representations of a posterior.
- (+) they distinguish between different types of hierarchical VAE models as models of the visual system by comparing their predictions on learned features and noise correlations.

Comment:
- More explanations on the conceptual motivation and differences between the different models proposed, especially between the bottom up vs top-down posterior factorization approaches, would have been useful to better understand the motivation for comparing these specific models (and not alternatives), and to understand the implications of the study for our understanding of visual processing.

Typo:
- I believe equation in line 72 is missing a term and should probably be p_theta(x|z_1,z_2) = p_theta(x|z_1)p_theta(z_1|z_2)

---

> ### Author Response · Authors · 2022-12-23
> **response for reviewer rFXb**
>
> We would like to thank the reviewer for their interest in our paper and for spotting a typo in our manuscript (which we fixed in the camera-ready version).
> Here are some explanations on the different models studied in the paper. In our manuscript, we took the one-layer LinearVAE baseline which is similar to the linear generative models of V1. In the two-layer graphical models, the conditional structure of the generative part can be either general (SkipVAE) or Markovian (TDVAE).
> The mathematical motivation of the recognition models (which we call top-down and bottom-up) is that they mimic the statistical properties of the true posterior. A more detailed explanation of this can be found in the Appendix (A.1.1). In the camera-ready version, we revised this section to make it more comprehensible and placed a reference to it in the main text.